# Aerosol Effects on the Development of Cumulus Clouds over the Tibetan Plateau

Xu Zhou[1,5], Naifang Bei[2], Hongli Liu[3], Junji Cao[1], Li Xing[1], Wenfang Lei[4], Luisa T. Molina[4], and Guohui Li[1*]

[1]Key Lab of Aerosol Chemistry and Physics, SKLLQG, Institute of Earth Environment, Chinese Academy of Sciences, Xi'an, China
[2]School of Human Settlements and Civil Engineering, Xi'an Jiaotong University, Xi'an, Shaanxi, China
[3]State Key Laboratory of Severe Weather, Chinese Academy of Meteorological Sciences, Beijing, China
[4]Molina Center for Energy and the Environment, La Jolla, CA, USA
[5]University of Chinese Academy of Science, Beijing, China
[*]Correspondence to: Guohui Li (ligh@ieecas.cn)

**Abstract.** The aerosol-cloud interaction over the Tibetan Plateau has been investigated using a cloud-resolving weather research and forecasting model with a two-moment bulk microphysical scheme including aerosol effects on cloud condensation nuclei and ice nuclei. Two types of cumulus clouds with a similar convective available potential energy, occurring over the Tibetan Plateau (Cu-TP) and North China Plain (Cu-NCP) in August 2014, are simulated to explore the response of convective clouds to aerosols. A set of aerosol profiles is used in the simulations, with the surface aerosol number concentration varying from 20 to 9000 $cm^{-3}$ and the sulfate mass concentration varying from 0.02 to 9.0 $\mu g\ cm^{-3}$. Increasing aerosol concentrations generally enhances the cloud core updraft and maximum updraft, intensifying convections in Cu-TP and Cu-NCP. However, the core updraft is much stronger in Cu-TP than Cu-NCP, because of the early occurrence of the glaciation process in Cu-TP that is triggered at an elevation above 4000 m. The precipitation increases steadily with aerosol concentrations in Cu-NCP, caused by the suppression of the warm rain but efficient mix-phased precipitation due to the reduced cloud droplet size. The precipitation in Cu-TP also increases with aerosol concentrations, but the precipitation enhancement is not substantial compared to that in Cu-NCP with high aerosol concentrations. The aerosol-induced intensification of convections in Cu-TP not only facilitates the precipitation, but also transports more ice-phase hydrometeors into the upper troposphere to decrease the precipitation efficiency. Considering the very clean atmosphere over the Tibetan Plateau, elevated aerosol concentrations can remarkably enhance convections due to its specific topography, which not only warms the middle troposphere to influence the Asian summer monsoon, but also delivers hydrometeors into the upper troposphere to allow more water vapor to travel into the lower stratosphere.

## 1    Introduction

Atmospheric aerosols, formed naturally and anthropogenically, influence the radiative energy budget of the Earth-atmosphere system in many ways. They scatter or absorb a fraction of the incoming solar radiation to cool or warm the atmosphere, decreasing surface temperature and altering atmospheric stability (e.g., Jacobson, 2002; Wang et al., 2013). They also serve as cloud condensation nuclei (CCN) and ice nuclei (IN), modifying optical properties and lifetime of clouds (e.g., Penner et al., 2001; Zhang et al., 2007). The aerosol indirect effect, generally referred to as the aerosol impact on cloud reflective properties and lifetime (Twomey, 1977; Houghton, 2001), has constituted one of the largest uncertainties in climate prediction (IPCC, 2013). In addition, the aerosol effects on precipitation have been regarded as an important but poorly understood process that could have major implications to climate and water supplies (Levin and Cotton, 2007; Wang et al., 2014a; b).

For a given amount of condensable water vapor, elevated aerosol concentrations increase the number of cloud droplets and reduce their sizes, enhancing not only the reflective properties but also the lifetime of clouds through suppressing warm rain processes (Twomey, 1977; Albrecht, 1989). Accumulative observational and modeling evidence has shown that reduced cloud droplet size, due to increasing CCN, inhibits collision and coalescence processes, suppressing warm rain and delaying the onset of precipitation. Therefore, more droplets are further allowed to be transported above the 0°C isotherm, triggering the efficient mixed-phase process to release more latent heat and intensify the convection (e.g. Rosenfeld and Lensky, 1998; Rosenfeld and Woodley, 2000; Kaufman and Nakajima, 1993; Andreae et al., 2004; Kaufman et al., 2005; Fan et al., 2007; Khain et al., 2008; Koren et al., 2010; Li et al., 2013; Loftus and Cotton, 2014). However, recent studies have shown that an optimal aerosol loading exists to invigorate convection (Rosenfeld et al., 2008; Koren et al., 2014; Dagan et al., 2015). Additionally, the aerosol impacts on cloud

developments are also proposed to be dependent on the environmental conditions, such as
relative humidity and vertical wind shear (van den Heever et al., 2007; Lee et al., 2008; Fan
et al., 2009; Tao et al., 2012; Fan et al., 2016).

66        The observational and model-derived evidence on how aerosols influence rainfall

remains elusive due to the complexity of cloud processes, which are determined by intricate
thermodynamic, dynamical, and microphysical processes and their interactions (Levin and
Cotton, 2007; McComiskey and Feingold, 2012; Lin et al., 2016). Observations have
demonstrated that the aerosol effect on precipitation depends on both the type of aerosols and
precipitating environments. Rainfall reduction has been observed in polluted industrial and
urban regions in shallow clouds or clouds with the top temperature exceeding -10°C (e.g.,
Rosenfeld, 2000; Ramanathan et al., 2001; Andreae et al., 2004; Yang et al., 2013). However,
documented rainfall increase has also been observed around heavily polluted coastal areas or
over oceans influenced by anthropogenic aerosols (e.g., Cerveny and Balling, 1998; Shepherd
and Burian, 2003; Zhang et al., 2007; Li et al., 2008b; Koren et al., 2012, 2014). Model
results tend to support the argument that increasing aerosol concentrations enhances
precipitation under a moist, unstable atmosphere (e.g., Khain et al., 2005; Fan et al., 2007; Li
et al., 2008a, 2009; Wang et al., 2011; Fan et al., 2013).

80        The Tibetan Plateau (TP), located in the central eastern Eurasia and with an average

elevation of more than 4000 m, significantly affects the formation and variability of the Asian
summer monsoon through mechanical and thermal dynamical effects (Wu et al., 2007). Due
to its strong surface heating, the cumulus clouds are active over the TP and can be organized
to form convective systems, contributing substantially to the precipitation over TP and
adjacent areas. The TP is surrounded by several important natural and anthropogenic aerosol
sources, and the in-situ and satellite measurements have shown that anthropogenic aerosols
and dust have been lofted to the TP, directly influencing the regional climate (Engling et al.,
2011). Soot aerosols deposited on the TP glaciers have been confirmed to contribute
significantly to observed glacier retreat (Xu et al., 2009). Absorbing aerosols over the TP
have been proposed to directly affect monsoon rainfall through the elevated heat pump
mechanism (Lau et al., 2008; D'Errico et al., 2015; Li et al., 2016).
However, to date few studies have been performed to investigate the aerosol indirect
effect or the aerosol-cloud interaction over the TP. In the present study, we report an
investigation of the aerosol effect on the cumulus cloud development and precipitation over
the TP. Two types of cumulus clouds occurring over the TP and the North China Plain (NCP)
are simulated using a cloud-resolving weather research and forecasting model for
comparisons. The model configuration is described in Section 2. The results and discussions
are presented in Section 3, and summary and conclusions are given in Section 4.

**2      Models and Design of Numerical Experiments**
**2.1    Model Configuration**
A cloud-resolving weather research and forecasting (CR-WRF) model (Skamarock et
al., 2004) is used in the study to simulate cumulus clouds. A two-moment bulk microphysical
scheme developed by Li et al. (2008a) is utilized to account for the aerosol-cloud interactions
in the simulations. The mass mixing ratio and number concentration of five hydrometeors are
predicted in the bulk microphysical scheme, including cloud water, rain water, ice crystal,
snow flake, and graupel. The gamma function is used to represent the size distribution of the
five hydrometeors. Detailed information is provided in Li et al. (2008a).
In order to consider the aerosol activation to CCN and IN, the CMAQ/models3 aerosol
module (Binkowski and Roselle, 2003) is implemented into the CR-WRF model. Aerosols
are simulated in the CMAQ using a modal approach assuming that particles are represented
by three superimposed log-normal size distributions. The aerosol species, including sulfate,
nitrate, ammonium, organic and black carbon, and other unidentified species (dust-like) are
predicted in the module.
For the CCN nucleation, the critical radius of dry aerosols is calculated from the
*k*-Köhler theory developed by Petters and Kreidenweis (2007; 2008; 2013) using water
supersaturation predicted by the CR-WRF model (Roger and Yau, 1989; Pruppacher and
Klett, 1997). If the activated CCN radius is less than 0.03 μm, the mass of water
condensation on CCN is calculated under the equilibrium assumption; otherwise, the mass of
water condensing on CCN is calculated by $m_w = K\frac{4}{3}\pi r_a^3 \rho_w$ at zero supersaturation, where
$3 < K < 8$ (Khain et al., 2000). Additionally, a novel, flexible approach, proposed by Philips
et al. (2008, 2013) has been used to parameterize the ice heterogeneous nucleation within
clouds. The method has empirically derived dependencies on the chemistry and surface area
of multiple species of IN aerosols, mainly including dust, black and organic carbon aerosols.
Three kinds of ice nucleation mechanisms are considered in the method, including contact,
immersion, and condensation freezing.
**2.2   Design of Numerical Experiments and Statistical Method in Data Analysis**
The spatial resolution used in the cloud simulations is 1 km in the horizontal direction
and about 250 m in the vertical direction. The model domain of $200 \times 200 \times 80$ grid boxes
along the x, y, and z directions, respectively, has been used to provide 200 km × 200 km
horizontal and 20-km vertical coverage in this study. The initial and boundary conditions of
water vapor are from the sounding data. The simulations use the open boundary conditions
under which variables of all horizontal gradients are zero at the lateral boundary.
Two types of cumulus clouds are simulated using the CR-WRF model. The cumulus
cloud over the TP (hereafter referred to as Cu-TP) is initialized using the sounding data
(87.08°E, 28.63°N, 4302 m a.s.l.) at 0800 UTC on August 24, 2014 (Figure 1a). The cumulus
cloud over the NCP (hereafter referred to as Cu-NCP) is initialized using the sounding data
(114.35°E, 37.17°N, 181 m a.s.l.) at 0800 UTC on August 12, 2014 (Figure 1b). The selected
sounding profiles over the TP and NCP reveal a moderate instability in the atmosphere, with
similar convective available potential energy (CAPE) for comparison, i.e., 675 J kg$^{-1}$ for
Cu-TP and 651 J kg$^{-1}$ for Cu-NCP. Although Cu-TP and Cu-NCP have the similar CAPE, the
remarkable difference of the initialization elevation between Cu-TP and Cu-NCP causes their
distinct development processes. The 0°C isotherm is generally at the level of around 5 km
a.s.l. in the summer. Therefore, when an air parcel perturbed in the boundary layer ascends to
form a cloud, the rising distance to the 0°C isotherm is around 1 km over the TP and about 4
km over the NCP. Therefore, the occurrence of the efficient mixed phase process is much
earlier for the cumulus cloud over the TP than the NCP, which substantially advances the
development of the cloud over the TP.
The cumulus development is triggered by a warm bubble 15-km wide and a maximum
temperature anomaly of 4°C at the height of 1.5 km a.g.l. (Li et al., 2008a) and the integration
time is two hours. Observed aerosol concentrations over the TP exhibit a large variation
during the monsoon season, i.e., the observed sulfate concentrations range from 0.1 to several
$\mu$g m$^{-3}$ (Decesari et al., 2010). Therefore, a set of 28 initial aerosol size distributions with the
aerosol number concentration ranging from 20 to 9000 cm$^{-3}$ and the sulfate mass
concentration ranging from 0.02 to 9.0 $\mu$g cm$^{-3}$ at the surface level are used. Other aerosol
species are scaled using the measurement at the Nepal Climate Observatory-Pyramid (NCO-P)
(Decesari et al., 2010). These aerosol distributions are designated for environments ranging
from very clean background air mass to polluted urban plumes over the TP and NCP.
Although the observed organic aerosol dominates the aerosol composition at NCO-P
(Decesari et al., 2010), considering the large uncertainties in the hygroscopicity of organic
aerosols, the hygroscopicity parameter for the secondary organic aerosol is set to be 0.05 in
the study (Petters and Kreidenweis, 2007; 2008). Hence, sulfate aerosols (or inorganic
aerosols) still play a dominant role in the CCN activation. The aerosol concentration is
assumed to decrease exponentially with height in the model simulations (Li et al., 2008a).
We have adopted several assumptions and simplifications for the processes associated
with aerosols. In the simulations, only the accumulation mode of aerosols is used for the
CCN and IN activation, and the aerosol spatial distributions are determined by the initial and
boundary conditions, without consideration of chemistry, emissions, and release from cloud
droplet evaporation or ice crystal sublimation. The sulfate, nitrate, ammonium, black carbon,
organic, and dust-like aerosols in the accumulation mode are included to consider the aerosol
CCN and IN effects. Therefore, the surface-level aerosol number concentration ([$Na$]) is used
to represent all types of aerosols, and the CCN concentration at a certain supersaturation (SS)
is not used in the study. It is worth noting that the simple aerosol assumption is subject to
cause rather large uncertainties in the aerosol activation to CCN and IN. Aerosol chemistry in
clouds plays a considerable role in the aerosol nucleation and growth. Direct emissions from
anthropogenic sources contribute substantially to the CCN and IN, even over the TP with
increasing human activities. Furthermore, mineral dust from the natural source frequently
dominates the TP throughout the year. Therefore, future studies need to be conducted to
include all the aerosol modes, chemistry, and emissions.
In order to evaluate the overall response of simulated cumulus clouds to changes in
aerosol concentrations, the population mean (p-mean) of a given variable over all qualified
grid points and for a given integration interval is used in the study (Wang, 2005), defined as:
$$\bar{C}^p = \frac{1}{\sum_{t=T_1}^{T_2} N(t)} \sum_{t=T_1}^{T_2} \sum_{n>n_{min}}^{q>q_{min}} c(x,y,t)$$
where $c$ represents a given quantity. The calculation only applies to the grid points where
both the mass concentration $q$ and number concentration $n$ of a hydrometeor or the
summation of several hydrometeors exceed the given minima. The total number of the grid
points at a given output time step $t$ is represented by $N(t)$. $T_1$ and $T_2$ are the start and end
output time steps, respectively.

**3.    Results and Discussions**
**3.1    Response of Cloud Properties to Changes in Aerosol Concentrations**
Figure 2a depicts the dependence of the p-mean of the cloud droplet number
concentration (CDNC) on the [$Na$]. Increasing [$Na$] provides more CCN to activate, and
although more activated droplets compete for the available water vapor, the water vapor
condensation efficiency is enhanced due to the increased bulk droplet surface area,
accelerating the latent heat release and the updraft to provide more supersaturated water
vapor. Therefore, the increasing CDNC is well consistent with increasing [$Na$] in Cu-TP and
Cu-NCP, in good agreement with previous studies (e.g., Fan et al., 2007a, b; Li et al., 2008a).
When the [$Na$] increases from about 20 cm$^{-3}$ to 9000 cm$^{-3}$, the p-mean of the CDNC increases
from 0.56 cm$^{-3}$ to 218 cm$^{-3}$ for Cu-NCP. However, more aerosols are activated in Cu-TP
compared to Cu-NCP, and the p-mean of the CDNC increases from 0.80 cm$^{-3}$ to 415 cm$^{-3}$ for
Cu-TP. Although the CAPE is similar for Cu-TP and Cu-NCP, the p-mean of CDNC in Cu-TP
is higher than that in Cu-NCP with the same [$Na$].
With the [$Na$] increasing from 20 to 9000 cm$^{-3}$, the effective radius of cloud droplet
($R_{eff}$) in Cu-TP is reduced from about 18.5 to 4 .1 μm, and the $R_{eff}$ in Cu-NCP is also
consistently reduced from 14.3 to 6.6 μm (Figure 2b). Interestingly, when the [$Na$] is less
than about 240 cm$^{-3}$, the $R_{eff}$ in Cu-TP is larger than that in Cu-NCP with the same [$Na$],
although the CDNC in Cu-TP is higher than that in Cu-NCP, showing more cloud water
condensed in Cu-TP. Figure 3a presents the dependence of the cloud water content (CWC) on
the [$Na$] in Cu-TP and Cu-NCP, showing that the CWC increases with increasing [$Na$]. This
positive relationship is caused by the combined effects of the increase in CDNC and the
decrease in $R_{eff}$, which inhibit the collision/coalescence of cloud droplets and also enhance
the water vapor condensation efficiency and the updraft to generate more available
condensable water vapor. The CWC in Cu-TP is higher than that in Cu-NCP for the same
[$Na$], due to higher CDNC and likely stronger updrafts in Cu-TP. The Cu-TP is triggered at
an elevation of more than 4000 m a.s.l. Therefore, considering that the 0°C isotherm is at the
level of around 5000 m a.s.l., the cloud water formed in the cumulus tends to be transported
above the 0°C isotherm to become supercooled, initiating the efficient mixed phase process
to release more latent heat and enhance the updraft. Therefore, there exists more supercooled
cloud water in Cu-TP than Cu-NCP when [$Na$] are same (Figure 3b).
Figure 4 provides the vertical profiles of the hydrometeors mass concentrations
(summed over the horizontal domain and then averaged during the simulation period) under
three aerosol scenarios: a very low [$Na$] of 90 cm$^{-3}$, a low [$Na$] of 900 cm$^{-3}$, and a high [$Na$]
of 9000 cm$^{-3}$, corresponding the background, clean, and polluted atmosphere, respectively. In
Cu-TP and Cu-NCP, the CWC achieves the highest under the high [$Na$] case and the lowest
under the very low [$Na$] case (Figures 4a and 4b). A higher [$Na$] enhances CDNC and reduces
$R_{eff}$, suppressing the conversion from cloud water to rain water and sustaining more CWC in
the cloud. In Table 1, the initial formation time of rain water is delayed with the [$Na$] increase
in Cu-TP and Cu-NCP. The height of the maximum CWC slightly increases from the very
low to high [$Na$] conditions in Cu-TP and Cu-NTP, but the maximum CWC occurs at 6~8 km
a.s.l. in Cu-TP and 2~4 km a.s.l. in Cu-NCP. Therefore, for Cu-TP, most of cloud droplets are
above the 0°C isotherm (about 5 km a.s.l.) and supercooled.
The ice particles (ice + snow) generally reach the highest in the high [$Na$] and lowest in
the very low [$Na$], which is consistent with those of the CWC in Cu-TP and Cu-NCP (Figures
4e and 4f). In the present study, the homogeneous freezing and rime-splintering mechanisms
(DeMott et al., 1994; Hallett and Mossop, 1974) are included for the ice nucleation. In

addition, the heterogeneous ice nucleation, including the contact, immersion, and condensation freezing, are all parameterized using the method proposed by Philips et al. (2008; 2013), and has considered the IN effect, depending not only on temperature and ice supersaturation, but also on the chemistry and surface area of multiple species of IN aerosols. The [Na] Enhancement generally suppresses the warm rain process to reduce the rain water, but provides more IN and supercooled CWC to accelerate the ice nucleation process. In addition, the rime-splintering mechanism also affects the ice particle profiles at the height with temperature ranging from -8°C and -3°C (Hallet and Mosssop, 1974). At the height of 6~8 km a.s.l. in Cu-TP and 4~6 km a.s.l. in Cu-NCP, the ice particles profiles are similar in the very low and low [Na] cases, which is caused by the rime-splintering mechanism. The ice crystal production from the rime-splintering mechanism is related to the graupel particles and the cloud droplets with radii exceeding 24 $\mu m$. Large cloud droplets in the very low [Na] facilitate the ice crystal productions from the rime-splintering mechanism, increasing the ice particles mass concentrations at the height of 6~8 km a.s.l. in Cu-TP and 4~6 km a.s.l. in Cu-NCP. Furthermore, there are more ice particles in Cu-TP than Cu-NCP with the same [Na] condition. The initial formation time of ice crystals is advanced by at least 12 minutes in Cu-TP compared to Cu-NCP (Table 1). The 0°C isotherm is at the level of around 5 km a.s.l. for the Cu-TP and Cu-NCP. However, the occurrence heights for the Cu-TP and Cu-NCP are more than 4 km and about 0.2 km a.s.l, respectively, and when an air parcel perturbed in the boundary layer ascends to form a cloud, the rising distance to the 0°C isotherm is about 1 km over the TP and around 4 km over the NCP. Therefore, the ice crystal formation time is significantly shortened in the Cu-TP compared to the Cu-NCP. The early formation of ice crystals not only facilitates their growth, also advances the glaciation process to intensify convections, further enhancing the growth process.

The rainwater in Cu-TP achieves the highest in the very low [$Na$] and lowest in the
high [$Na$], and vice versa in Cu-NCP (Figures 4c and 4d). If not considering the contribution
of graupel melting to the rainwater, enhancement of [$Na$] suppresses the warm rain process to
reduce the rainwater, but enhances the raindrop size, which conversely accelerates the
raindrop falling (Table 1). In Cu-TP, due to relatively low temperature below the freezing
level and short falling distance (about 1 km), graupels dominate the precipitating particles,
melting less to rainwater. So early occurrence of the warm rain process in the very low [$Na$]
case causes the most rainwater formation (Figure 4c). However, in Cu-NCP, graupels falling
below the freezing level tend to melt due to high temperature and long falling distance (about
4 ~ 5 km), enhancing the rainwater formation. More ice particles and supercooled CWC in
the high [$Na$] case are favorable for the ice growth through deposition, aggregation among ice
crystals, and riming of supercooled droplets (Wang and Change, 1993a, b; Lou et al., 2003),
and heavily rimed ice crystals are transferred to graupels, enhancing the graupel formation.
Therefore, in Cu-NCP, the high [$Na$] corresponds to the maximum graupel content and also
rainwater content (Figures 4d and 4h). However, in Cu-TP, below 12 km, the low [$Na$]
corresponds to the largest amounts of graupels. Early occurrence of the glaciation process in
Cu-TP causes most of raindrops to be frozen to form graupels. The freezing rate of raindrops
depends on the temperature, the raindrop size and number, and their corresponding variations
with time (Lou et al., 2003). Generally, the raindrops with the larger size are easier to be
frozen under the lower temperature. The [$Na$] Enhancement decreases the raindrop number,
but increases its size and updraft to lower the temperature, causing the maximum raindrop
freezing efficiency under the low [$Na$] condition. In addition, increasing the [Na] invigorates
the convection and produce larger graupels, and then the melting of the graupel causes the
formation of larger raindrops (Table 1).
It is worth noting that ice particles and graupels are transported above 12 km a.s.l. or
even exceeding 16 km a.s.l. (near tropopause) in Cu-TP, showing intensified convection and
also contributing to moistening the upper troposphere.
**3.2    Response of Convective Strength to Changes in Aerosol Concentrations**
The p-mean of the updraft and downdraft in a core area is used to measure the
convective strength of the simulated cumulus clouds, which is defined by the absolute
vertical wind speed exceeding 1 m s$^{-1}$ and total condensed water mixing ratio more than 10$^{-2}$
g kg$^{-1}$ (*Wang*, 2005). When the [*Na*] increases from 20 to 9000 cm$^{-3}$, the p-mean of the core
updraft increases from 2.0 to 4.3 m s$^{-1}$ in Cu-TP, and from 1.5 to 2.7 m s$^{-1}$ in Cu-NCP (Figure
5a). The enhancement of the core updraft with increasing [*Na*] is caused by the suppression
of the warm rain process to induce the more efficient mixed phase process, releasing more
latent heat to intensify the convection. With the same [*Na*], the p-mean of the core updraft is
larger in Cu-TP than in Cu-NCP, showing the significant impact of the early occurrence of
the glaciation process on the cloud development.
In Cu-TP, with the [*Na*] increase, the p-mean of the downdraft increases when the [*Na*]
is less than 90 cm$^{-3}$, but it becomes insensitive to the changes in [*Na*] when the [*Na*] is
between 90 and 1800 cm$^{-3}$, and commences to decrease when the [*Na*] exceeds 1800 cm$^{-3}$
(Figure 5b). The complex nonlinear variation of the p-mean of the downdraft with the [*Na*]
reflects the change in the vertical distribution of ice particles and graupels caused by the
enhancement of [*Na*] in Cu-TP. The enhancement of the convective strength with increasing
[*Na*] not only intensifies the convection to facilitate precipitation, producing more
precipitable particles, but also transports more ice particles and graupels to the upper
troposphere due to the specific topography and further suppress the occurrence of the
downdraft. However, the p-mean of the downdraft in Cu-NCP increases steadily with [*Na*].
Such an increase in the core downdraft with [*Na*] might be caused by the formation of a large
mass loading of precipitable particles to reduce buoyancy and increase downdrafts.
Interestingly, when the [$Na$] is less than about 450 cm$^{-3}$, the p-mean of downdraft in Cu-TP is
greater than that in Cu-NCP, but opposite when [$Na$] exceeding 450 cm$^{-3}$, indicating the
influence of the early occurrence of the glaciation process due to the specific topography in
Cu-TP.
The maximum updraft, representing the largest local latent heat release, generally
increases with [$Na$] in Cu-TP and Cu-NCP (Figure 6a). The maximum updraft in Cu-TP is
much higher than that in Cu-NCP with the same [$Na$]. In Cu-TP, when the [$Na$] exceeds 750
cm$^{-3}$, the maximum updraft becomes insensitive to changes in the [$Na$]. In Cu-NCP, the
maximum updraft is not very sensitive to changes in the [$Na$] when the [$Na$] exceeds 2400
cm$^{-3}$. The maximum downdraft, or the largest drag speed, indicating the largest strength to
inhibit the development of the convection, also increases generally with the [$Na$] in Cu-TP
and Cu-NCP (Figure 6b), but Cu-TP produces the more intensive maximum downdraft than
Cu-NCP.
**3.3    Response of Precipitation to Changes in Aerosol Concentrations**
Figure 7 shows the variation of the accumulated precipitation with [$Na$] in Cu-TP and
Cu-NCP. Generally, the precipitation increases with [$Na$], which is consistent with previous
modeling studies (e.g., Khain et al., 2005, 2008; Fan et al., 2007; Li et al., 2008a; 2009).
Since Cu-TP and Cu-NCP occur under humid conditions, the precipitation enhancement with
[$Na$] is also in good agreement with measurements. Observations have shown the
precipitation enhancement around heavily polluted coastal urban areas (Shepherd and Burian,
2003; Ohashi and kida, 2002) or over oceans influenced by pollution aerosols (Cerveny and
Balling, 1998; Li et al., 2008b; Koren et al., 2012, 2014).
When the [$Na$] is increased from about 20 cm$^{-3}$ to 9000 cm$^{-3}$, the precipitation of
Cu-TP increases from 0.13 mm to 0.23 mm; when the [$Na$] exceeds 300 cm$^{-3}$, the

precipitation becomes insensitive to the variation in [$Na$]. In contrast, the precipitation of Cu-NCP consistently increases from 0.03 mm to 0.37 mm with [$Na$] ranging from 20 cm$^{-3}$ to 9000 cm$^{-3}$. In addition, when the [$Na$] is less than 500 cm$^{-3}$, Cu-TP produces more precipitation than Cu-NCP, which can be explained by the early occurrence of the glaciation process causing less warm rain but more efficient mixed-phase processes. However, when the [$Na$] exceeds 500 cm$^{-3}$, the precipitation efficiency of Cu-NCP is higher than that of Cu-TP, although the convective strength is larger in Cu-TP than Cu-NCP. The increasing convective strength with [$Na$] not only enhances the precipitation, but also transports more ice and graupel particles above 12 km to form the anvil. The ice particles and grauples in the anvil are subject to sublimation and evaporation to moisten the upper troposphere, and decrease the precipitation efficiency in Cu-TP.

The water content and precipitation in the Cu-TP response well monotonically to the changes in the [$Na$]. Numerous studies have shown that the reduced liquid water path (LWP) by increasing aerosols under relatively dry conditions (e.g., Khain et al., 2005). During the summer monsoon season, the atmosphere over the TP is humid due to the water vapor transport by the monsoon (Figure 1a). The ambient humidity in the simulations of the Cu-TP exceeds 80% in the low-level atmosphere, causing the good monotonicity in the responses of water content and precipitation to aerosols.

### 3.4  Sensitivity Studies

Recent studies have demonstrated that convection is more active and stronger during summertime over Tibetan Plateau due to its unique thermodynamic forcing (Hu et al., 2016). We have further performed sensitivity studies to explore the impact of the maximum perturbation temperature (MPT) in the warm bubble on the development of cumulus clouds. The MPTs of 2.0°C and 0.5°C are used to trigger Cu-TP and Cu-NCP with the [$Na$] ranging from 20 cm$^{-3}$ to 9000 cm$^{-3}$.

For Cu-TP, the core updraft decreases slightly when the MPT is reduced from 4.0°C to
2.0°C, particularly when the [$Na$] exceeds 100 cm$^{-3}$, the decrease of the core updraft is
indiscernible. When the MPT is reduced from 2.0°C to 0.5°C, the core updraft decreases
considerably. However, for Cu-NCP, the core updraft decreases substantially when the MPT
is reduced from 4.0°C to 0.5°C. When the MPT is 0.5°C and the [$Na$] is less than 80 cm$^{-3}$, the
updraft core area is not formed in Cu-NCP. When the MPT is the same, the core updraft is
much larger in Cu-TP than Cu-NCP with the same [$Na$]; even the core updraft in Cu-TP with
the MPT of 0.5°C is larger than that in Cu-NCP with the MPT of 4.0°C when the [$Na$] is
more than 80 cm$^{-3}$. Therefore, under the unstable conditions over the Tibetan Plateau, a small
perturbation can induce strong convections, which is primarily caused by early occurrence of
the glaciation process due to the specific topography, as discussed in Section 3.1.
The accumulated precipitation generally decreases with the MPT in Cu-TP and
Cu-NCP with the same [$Na$]. When the MPT is 4.0°C, Cu-NCP produces more precipitation
than Cu-TP with the [$Na$] exceeding 500 cm$^{-3}$, but Cu-TP produces much more precipitation
than Cu-NCP with the MPT of 0.5°C under all aerosol conditions. In addition, the
precipitation generally increases with increasing the [$Na$] in Cu-TP and Cu-NCP with various
MPTs, and does not exhibit a nonlinear variation with the [$Na$], which is not consistent with
the results in Li et al. (2008a). The possible reason is that in this study, the maximum p-mean
of CDNC is about 410 cm$^{-3}$, which is much less than that in Li et al. (2008a). If the [$Na$] is
further increased, the precipitation might be suppressed.

**4.    Summary and Conclusions**
The aerosol-cloud interaction over the TP has been examined using the CR-WRF
model with a two moment microphysical scheme considering the aerosol effects on CCN and
IN. For comparisons, two types of cumulus clouds, occurring over the TP and NCP in August
2014, are modeled to examine the response of the cumulus clouds development to the change
in aerosol concentrations. A set of 28 aerosol profiles are utilized in simulations, with the
surface aerosol number concentration varying from 20 to 9000 $cm^{-3}$ and the sulfate mass
concentration varying from 0.02 to 9.0 μg $cm^{-3}$. Multiple aerosol species are considered to
provide CCN and IN, including sulfate, nitrate, ammonium, organic and black carbon, and
dust-like aerosols.
In general, with varying aerosol concentrations from very clean background condition
to the polluted condition, more aerosols are activated, significantly increasing the CDNC and
decreasing the droplet size in Cu-TP and Cu-NCP. Formation of a large amount of cloud
droplets with small sizes suppresses the warm rain process and enhances water vapor
condensation efficiency and updraft to generate more available condensable water vapor.
When more cloud droplets are transported above the 0°C isotherm, occurrence of the
mixed-phase process releases more latent heat to further enhance the cloud core updraft and
increase precipitation, intensifying the convections in Cu-TP and Cu-NCP.
However, early occurrence of the glaciation process in Cu-TP, which is triggered at an
elevation of more than 4000 m, causes large differences between Cu-TP and Cu-NCP. Much
more supercooled cloud droplets are formed in Cu-TP than Cu-NCP with the same aerosol
concentration, facilitating the mixed-phase process and significantly enhancing the core
updraft and maximum updraft in Cu-TP compared to Cu-NCP. Nevertheless, the intensified
convection induced by the increase of aerosol concentrations in Cu-TP not only facilitates the
precipitation, but also delivers more ice-phase hydrometeors into the upper troposphere to
form the anvil, decreasing the precipitation efficiency. Therefore, in Cu-TP, when aerosol
concentrations are high, the precipitation enhancement becomes insignificant with increasing
aerosol concentrations, but a considerable amount of ice-phase hydrometeors are lofted above
12 km or even exceeding 16 km. Additionally, sensitivity studies have also shown that under
the unstable conditions over the TP, a small perturbation in temperature can induce strong
convections, which is primarily caused by early occurrence of the glaciation process due to
the specific topography.
In the present study, both CCN and IN effects are considered in the cloud simulations.
However, there are still difficulties in quantitatively distinguish those two effects on the
ice-phase cloud development using sensitivity studies. Obviously, the CCN plays a dominant
role in the mixed-phase cloud development. Even when the IN is scare in the atmosphere, the
mixed-phase cloud development is not hindered with sufficient CCN, because freezing of
raindrops, the subsequent splinter-riming process, and homogeneous freezing of cloud
droplets still initialize the glaciation process to facilitate the development of the mixed phase
cloud.
It is worth noting that, although the CAPE is similar for the Cu-TP and Cu-NCP, it
might not be fair to compare aerosol impacts on the cloud development over the TP with the
NCP, considering the difference of the water vapor profile, wind shear, topography, and
anthropogenic and natural aerosol sources between the two regions. However, the
comparisons have highlighted that the topography plays a large role in the development of
cumulus over the TP.
Rapid growth of industrialization, urbanization, and transportation in Asia has caused
severe air pollution, progressively increasing aerosol concentrations in the regions
surrounding TP. Pollution aerosols from surrounding areas have been observed to be
transported to the TP. Considering the very clean atmosphere over the TP, elevated aerosol
concentrations can considerably enhance the convections due to its specific topography.
Numerous studies have shown that the TP significantly influences the formation and
variability of the Asian summer monsoon through mechanical and thermal dynamical effects
(e.g., Wu et al., 2007). In addition, Fu et al. (2006) have reported that convection over the TP
provides the main pathway for cross-tropopause transport in the Asian monsoon/TP region.
Hence, intensification of convections due to the increase of aerosol concentrations over the
TP not only enhances the latent heat release to warm the middle troposphere, influencing the
Asian summer monsoon, also delivers more hydrometeors into the upper troposphere,
allowing more water vapor to travel into the lower stratosphere. Further studies are needed to
evaluate the aerosol indirect effect on the Asian summer monsoon and the
troposphere/stratosphere exchange over the TP.

*Acknowledgements*. This work was supported by the National Natural Science Foundation of
China (No. 41275153) and by the "Hundred Talents Program" of the Chinese Academy of
Sciences. Naifang Bei is supported by the National Natural Science Foundation of China (No.
41275101). Luisa Molina and Wenfang Lei acknowledge support from US NSF Award

447 1560494.

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

Table 1 Response of cloud properties in Cu-TP and Cu-NCP under three aerosol conditions[*].

| Clouds | Cu-TP | | | Cu-NCP | | |
|---|---|---|---|---|---|---|
| | Background | Clean | Polluted | Background | Clean | Polluted |
| Initial formation time of hydrometeors (minutes) | | | | | | |
| Rain | 10 | 14 | 20 | 8 | 10 | 14 |
| Ice crystal | 12 | 10 | 8 | 24 | 24 | 26 |
| Graupel | 12 | 14 | 16 | 18 | 18 | 16 |
| P-mean of effective radius of hydrometeors (μm) | | | | | | |
| Rain | 119 | 132 | 647 | 110 | 151 | 223 |
| Graupel | 559 | 665 | 917 | 221 | 303 | 447 |

[*]The aerosol concentrations are 90, 900, and 9000 $cm^{-3}$ for the background, clean, and polluted conditions,
respectively.

**Figure Captions**

Figure 1 Atmospheric sounding (a) over the Tibetan Plateau (87.08°E, 28.63°N, 4302 m a.s.l.) at 0800 UTC on August 12, 2014 and (b) over North China Plain (114.35°E, 37.17°N, 181 m a.s.l.) at 0800 UTC on August 24, 2014. The black line corresponds to the temperature, and the blue line represents the dew point temperature.

Figure 2 Modeled p-mean of (a) cloud droplet number concentration and (b) effective radius as a function of the initial $[N_a]$ in Cu-TP and Cu-NCP.

Figure 3 Modeled p-mean of (a) cloud water mass concentration and (b) supercooled cloud water mass concentration as a function of the initial $[N_a]$ in Cu-TP and Cu-NCP in Cu-TP and Cu-NCP.

Figure 4 Vertical profiles of time-averaged masses of hydrometeors under background (90 cm$^{-3}$, blue), clean (900 cm$^{-3}$, green), and polluted (9000 cm$^{-3}$, red) $[N_a]$ for (a) and (b) cloud water, (c) and (d) rain water, (e) and (f) ice particles (ice + snow), and (g) and (h) graupel in Cu-TP and Cu-NCP, respectively. The brown solid and dotted lines represent the surface level and the 0°C isotherm, respectively.

Figure 5 Simulated p-mean of (a) updraft and (b) downdraft in the core area (defined as an area where the absolute vertical velocity of wind is greater than 1 m s$^{-1}$ and the total condensed water content exceeds $10^{-2}$ g kg$^{-1}$) as a function of the initial $[N_a]$ in Cu-TP and Cu-NCP.

Figure 6 Modeled (a) maximum updraft and (b) minimum downdraft as a function of the initial $[N_a]$ in Cu-TP and Cu-NCP.

Figure 7 Modeled cumulative precipitation inside the model domain (mm) as a function of the initial $[N_a]$ in Cu-TP and Cu-NCP.

Figure 8 Response of (a) the p-mean of core updraft and (b) cumulative precipitation inside the model domain to the change in the maximum perturbation temperature of the warm bubble under various aerosol conditions in Cu-TP and Cu-NCP.

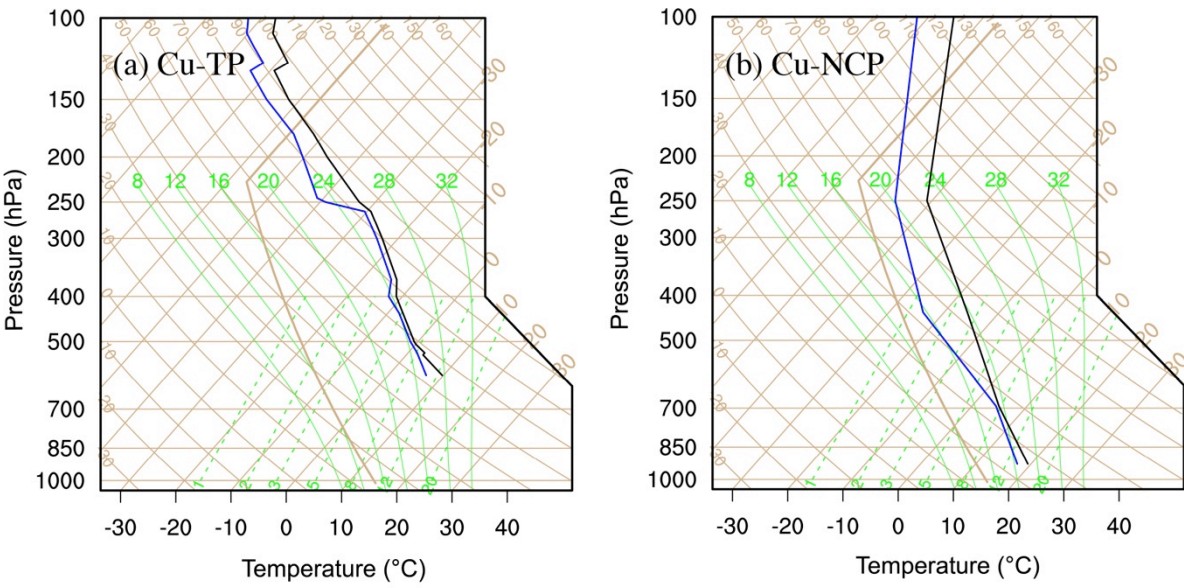

Figure 1 Atmospheric sounding (a) over the Tibetan Plateau (87.08°E, 28.63°N, 4302 m a.s.l.)
at 0800 UTC on August 12, 2014 and (b) over North China Plain (114.35°E, 37.17°N, 181 m
a.s.l.) at 0800 UTC on August 24, 2014. The black line corresponds to the temperature, and
the blue line represents the dew point temperature.

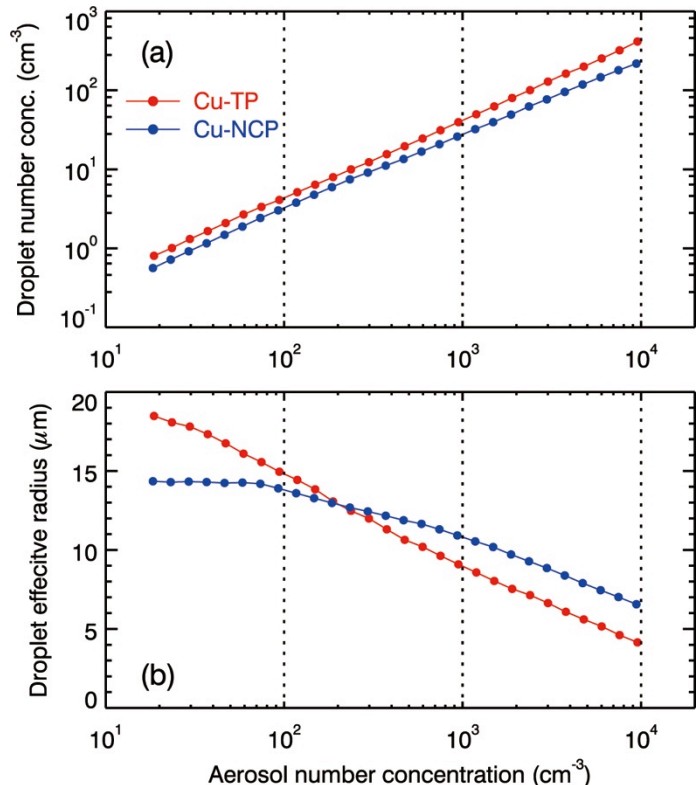

Figure 2 Modeled p-mean of (a) cloud droplet number concentration and (b) effective radius
as a function of the initial $[N_a]$ in Cu-TP and Cu-NCP.

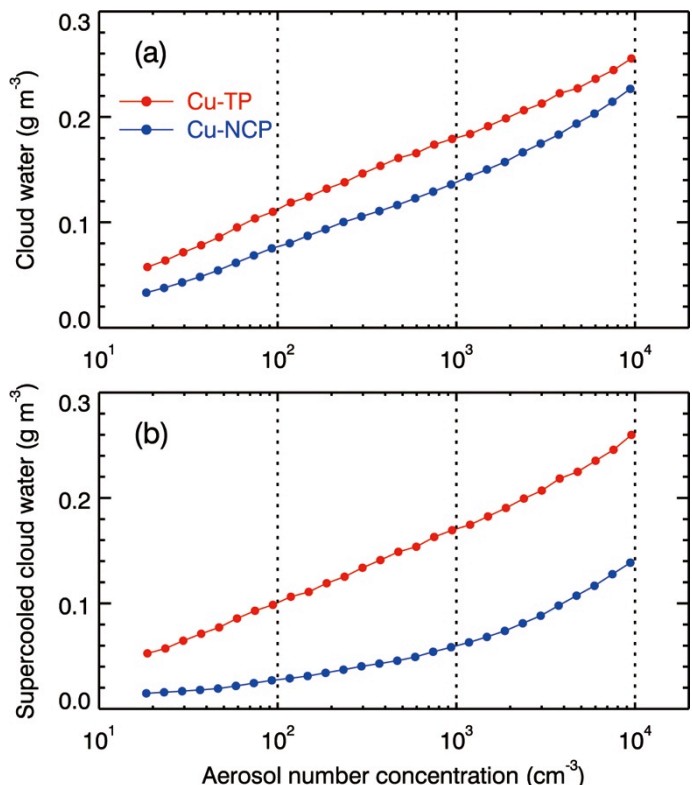

Figure 3 Modeled p-mean of (a) cloud water mass concentration and (b) supercooled cloud
water mass concentration as a function of the initial $[N_a]$ in Cu-TP and Cu-NCP in Cu-TP and
Cu-NCP.

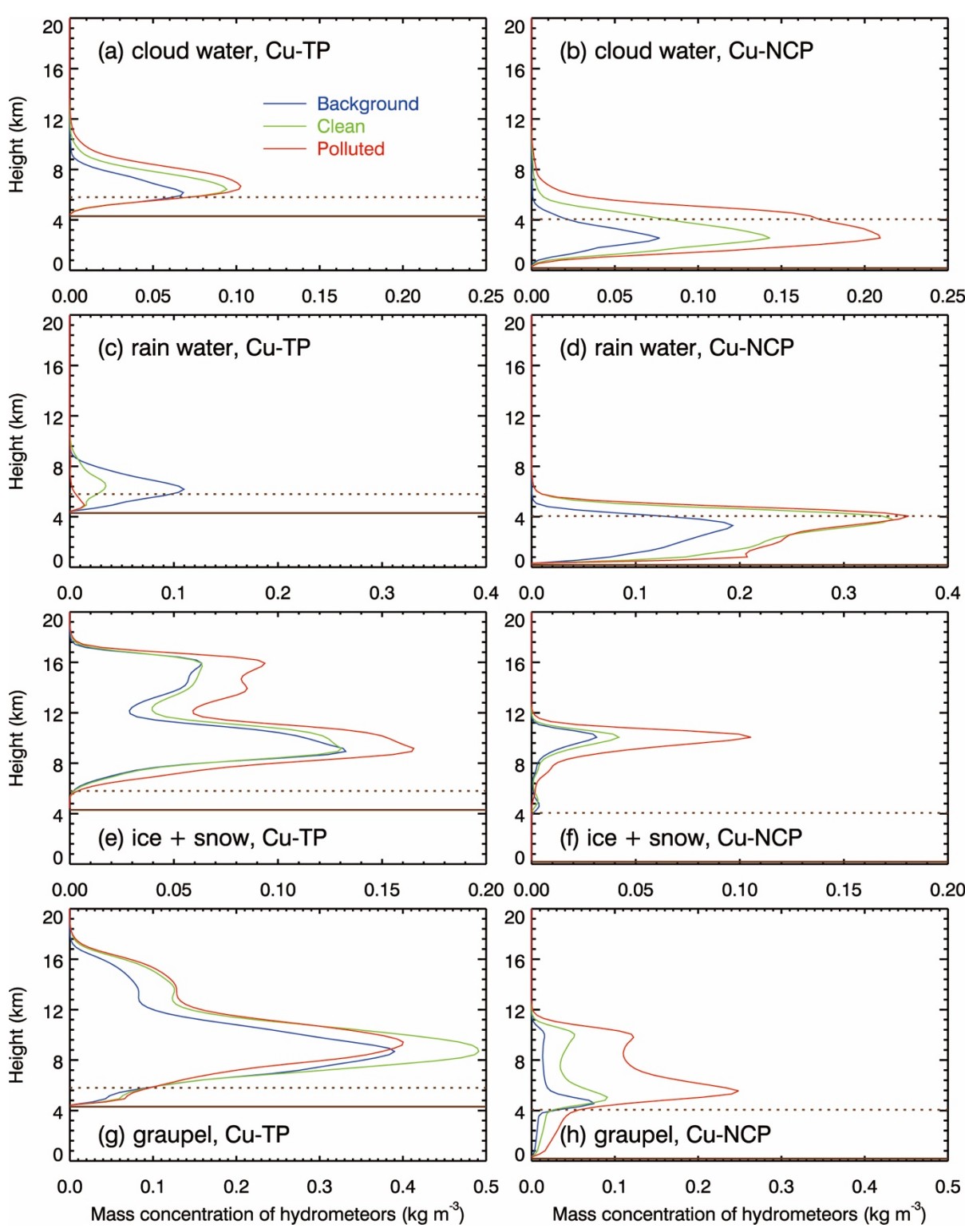

Figure 4 Vertical profiles of time-averaged masses of hydrometeors under background (90 cm$^{-3}$, blue), clean (900 cm$^{-3}$, green), and polluted (9000 cm$^{-3}$, red) [$N_a$] for (a) and (b) cloud water, (c) and (d) rain water, (e) and (f) ice particles (ice + snow), and (g) and (h) graupel in Cu-TP and Cu-NCP, respectively. The brown solid and dotted lines represent the surface level and the 0°C isotherm, respectively.

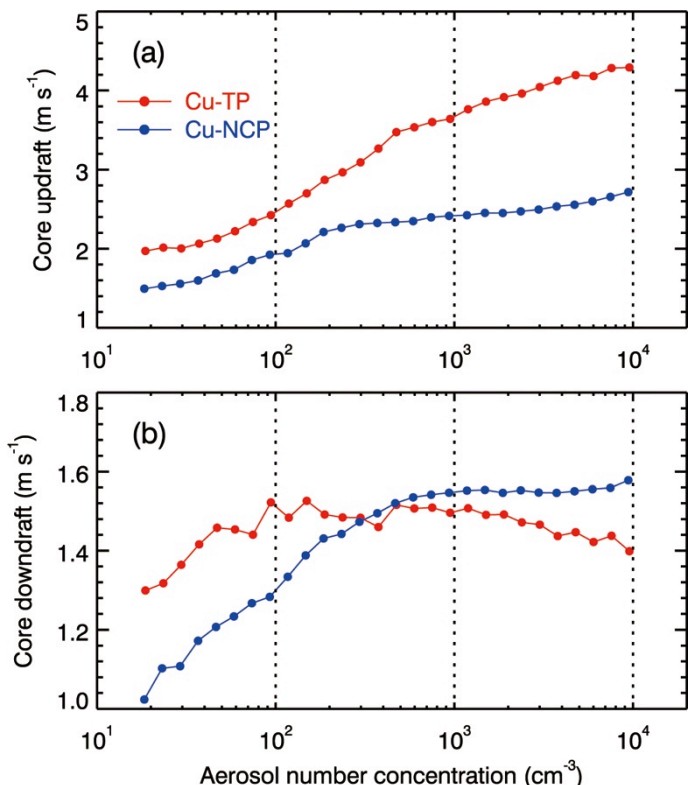

Figure 5 Simulated p-mean of (a) updraft and (b) downdraft in the core area (defined as an
area where the absolute vertical velocity of wind is greater than 1 m s$^{-1}$ and the total
condensed water content exceeds $10^{-2}$ g kg$^{-1}$) as a function of the initial [$N_a$] in Cu-TP and
Cu-NCP.

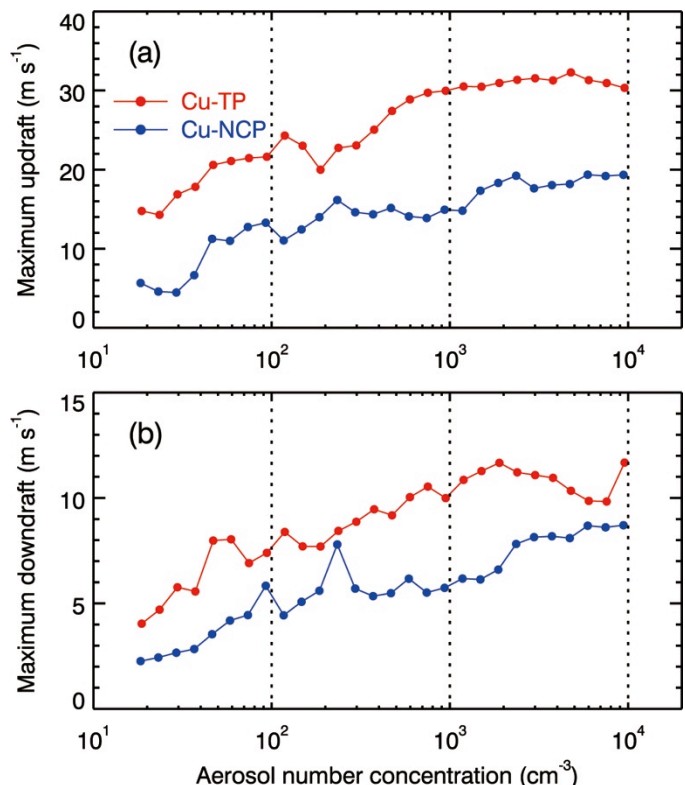

Figure 6 Modeled (a) maximum updraft and (b) minimum downdraft as a function of the
initial [$N_a$] in Cu-TP and Cu-NCP.

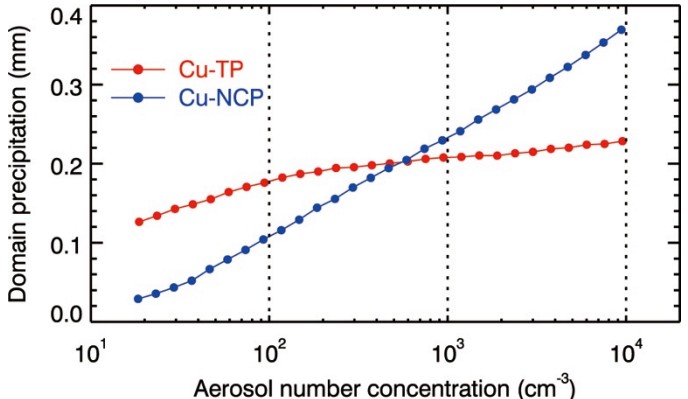

Figure 7 Modeled cumulative precipitation inside the model domain (mm) as a function of
the initial $[N_a]$ in Cu-TP and Cu-NCP.

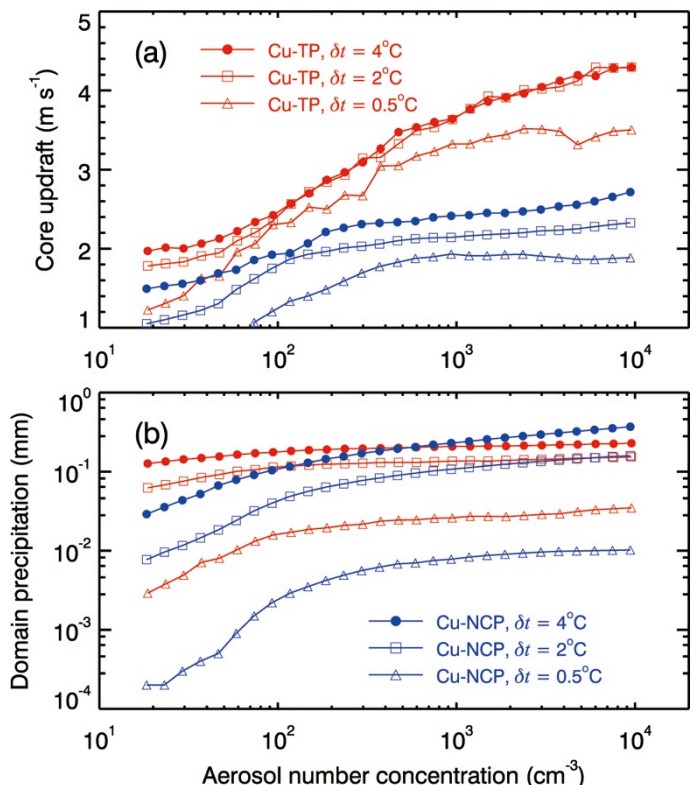

Figure 8 Response of (a) the p-mean of core updraft and (b) cumulative precipitation inside
the model domain to the change in the maximum perturbation temperature of the warm
bubble under various aerosol conditions in Cu-TP and Cu-NCP.