# Peer review of "Aerosol Effects on the Development of Cumulus Clouds over the Tibetan Plateau"

_Atmospheric Chemistry and Physics, 2017_

## Referee Comment (RC1) · Anonymous Referee #1 · 16 Mar 2017

This study uses a cloud-resolving model with an aerosol-aware cloud microphysics to investigate the convective cloud responses to aerosol perturbations over the Tibetan Plateau (TP). Considering the special topographic and meteorological conditions over TP, this study really extends the current aerosol-cloud interaction (ACI) research into a new regime on the earth. It is also interesting to see the comparison of the ACI between TP and the North China Plain for the similar type of cloud. The sensitivity experiments by perturbing the initial cloud convective strength demonstrate the simulated cloud and precipitation responses are robust. I have some minor comments for authors to address before I can recommend accepting this paper.

1. Both CCN and IN effects are considered in the CR-WRF cloud microphysics. Can authors distinguish those two effects on the mixed-phase clouds simulated in this study?

2. It is a little surprising to see the so good monotonicity in the responses of water content and precipitation to aerosols. Lots of studies have reported the reduced LWP by increasing aerosols under relative dry conditions. What is the ambient humidity in the simulations? How to treat moisture sources in the initial and boundary conditions? I would assume the water vapor amount can be limited over TP.

3. In Figure 4, please plot where is the surface level and where is the freezing level (0 degree isotherm). Those are very important information, as your later explanation of the differences of aerosol effects between TP and NCP relies on them. With the lower freezing level at TP, does it also mean less liquid water content and less room for aerosol invigoration effect?

4. L258, how does aerosol increase raindrop size and then foster freezing efficiency? I would think the other way, i.e., aerosols invigorate the convection, produce larger graupel, and then the melting of the graupel gives you larger raindrop.

5. Table 1, the initial formation times of hydrometeors are not fully discussed. Why ice crystal formations time is shorten by aerosols in TP but prolonged in NCP?

6. For the aerosol concentrations shown in each plot, do they represent all types of aerosols in all size modes? How about using CCN concentration at 0.1% SS instead?

7. As the clouds in this study are precipitating, it is better called them cumulonimbus, rather than cumulus.

---

## Referee Comment (RC2) · Anonymous Referee #2 · 26 Mar 2017

This study seeks to show the relationship between aerosol loading and deep convection within the Tibetan Plateau region and compare with the North China Plain. Model simulations are used to show the sensitivity of deep convection to various concentrations of aerosol loading. I believe this paper can be published after several grammatical errors and some technical questions are addressed. For most of the grammatical error revisions please see the attached PDF.

How confident are you in the aerosol assumptions used as input to your model? Aerosol chemistry is relevant when certain aerosols activate as CCN. Also, emission data is important since the Tibetan Plateau does not often witness heavy aerosol loading events, though you are correct about the summer season having the most polluted conditions. Also, coarse mode aerosols (mineral dust) tend to dominate that region throughout the year.

Do you expect to see much sulfate in the Tibetan Plateau? And if so, how does that compare to the ability of carbonaceous aerosols to activate as CCN since they too can be observed over this region.

I have no problems with the results you show from your model but both comparison regions should be equivalent. It seems that topography plays a large role in the development of deep convection over the Tibetan Plateau. I believe your point is that the monsoon may be affected by aerosol-cloud interactions over this region. However, is it a fair to compare aerosol impacts on storm development in the Tibetan Plateau region with the Northern China Plains where the topography is not only flatter but there are more sources of aerosol loading as well.

Please also note the supplement to this comment:
http://www.atmos-chem-phys-discuss.net/acp-2017-148/acp-2017-148-RC2-supplement.pdf

---

## Author Comment (AC1)

**Reply to Anonymous Referee #1**

We thank the reviewer for the careful reading of the manuscript and helpful comments. We have revised the manuscript following the suggestion, as described below.

This study uses a cloud-resolving model with an aerosol-aware cloud microphysics to investigate the convective cloud responses to aerosol perturbations over the Tibetan Plateau (TP). Considering the special topographic and meteorological conditions over TP, this study really extends the current aerosol-cloud interaction (ACI) research into a new regime on the earth. It is also interesting to see the comparison of the ACI between TP and the North China Plain for the similar type of cloud. The sensitivity experiments by perturbing the initial cloud convective strength demonstrate the simulated cloud and precipitation responses are robust. I have some minor comments for authors to address before I can recommend accepting this paper.

**1 Comment**: Both CCN and IN effects are considered in the CR-WRF cloud microphysics. Can authors distinguish those two effects on the mixed-phase clouds simulated in this study?

**Response:** We have clarified in Section 4: "*In the present study, both CCN and IN effects are considered in the cloud simulations. However, there are still difficulties in quantitatively distinguish those two effects on the ice-phase cloud development using sensitivity studies. Obviously, the CCN plays a dominant role in the mixed-phase cloud development. Even when the IN is scare in the atmosphere, the mixed-phase cloud development is not hindered with sufficient CCN, because freezing of raindrops, the subsequent splinter-riming process, and homogeneous freezing of cloud droplets still initialize the glaciation process to facilitate the development of the mixed phase cloud.*"

**2 Comment**: It is a little surprising to see the so good monotonicity in the responses of water content and precipitation to aerosols. Lots of studies have reported the reduced LWP by increasing aerosols under relative dry conditions. What is the ambient humidity in the simulations? How to treat moisture sources in the initial and boundary conditions? I would assume the water vapor amount can be limited over TP.

**Response:** We have clarified in Section 3: "*The water content and precipitation in the Cu-TP response well monotonically to the changes in the [Na]. Numerous studies have shown that the*

*reduced liquid water path (LWP) by increasing aerosols under relatively dry conditions (e.g., Khain et al., 2005). During the summer monsoon season, the atmosphere over TP is humid due to the water vapor transport by the monsoon (Figure 1a). The ambient humidity in the simulations of the Cu-TP exceeds 80% in the low-level atmosphere, causing the good monotonicity in the responses of water content and precipitation to aerosols.*". We have also clarified in Section 2: "*The initial and boundary conditions of water vapor are from the sounding data. The simulations use the open boundary conditions under which variables of all horizontal gradients are zero at the lateral boundary.*".

**3 Comment**: In Figure 4, please plot where is the surface level and where is the freezing level (0 degree isotherm). Those are very important information, as your later explanation of the differences of aerosol effects between TP and NCP relies on them. With the lower freezing level at TP, does it also mean less liquid water content and less room for aerosol invigoration effect?

**Response:** We have modified Figure 4 as suggested. The lower freezing level generally reduces the liquid water content but causes early occurrence of the glaciation process, enhancing the aerosol invigoration effect.

**4 Comment**: L258, how does aerosol increase raindrop size and then foster freezing efficiency? I would think the other way, i.e., aerosols invigorate the convection, produce larger graupel, and then the melting of the graupel gives you larger raindrop.

**Response:** We have clarified in Section 3: "*In addition, increasing the [Na] invigorates the convection and produce larger graupels, and then the melting of the graupel causes the formation of larger raindrops (Table 1).*"

**5 Comment**: Table 1, the initial formation times of hydrometeors are not fully discussed. Why ice crystal formations time is shortened by aerosols in TP but prolonged in NCP?

**Response:** We have clarified in Section 3: "*The 0°C isotherm is at the level of around 5 km a.s.l. for the Cu-TP and Cu-NCP. However, the occurrence heights for the Cu-TP and Cu-NCP are more than 4 km and about 0.2 km a.s.l, respectively, and when an air parcel perturbed in the boundary layer ascends to form a cloud, the rising distance to the 0°C isotherm is about 1*

*km over the TP and around 4 km over the NCP. Therefore, the ice crystal formation time is significantly shortened in the Cu-TP compared to the Cu-NCP.*"

**6 Comment**: For the aerosol concentrations shown in each plot, do they represent all types of aerosols in all size modes? How about using CCN concentration at 0.1% SS instead?

**Response:** We have clarified in Section 2: "*The sulfate, nitrate, ammonium, black carbon, organic, and dust-like aerosols in the accumulation mode are included to consider the aerosol CCN and IN effects. Therefore, the surface-level aerosol number concentration ([Na]) is used to represent all types of aerosols, and the CCN concentration at a certain supersaturation (SS) is not used in the study.*"

**7 Comment**: As the clouds in this study are precipitating, it is better called them cumulonimbus, rather than cumulus.

**Response:** We have changed "cumulus" to "cumulonimbus" when precipitation occurs in clouds.

---

## Author Comment (AC2)

**Reply to Anonymous Referee #2**

We thank the reviewer for the careful reading of the manuscript and helpful comments. We have revised the manuscript following the suggestion, as described below.

This study seeks to show the relationship between aerosol loading and deep convection within the Tibetan Plateau region and compare with the North China Plain. Model simulations are used to show the sensitivity of deep convection to various concentrations of aerosol loading. I believe this paper can be published after several grammatical errors and some technical questions are addressed. For most of the grammatical error revisions please see the attached PDF.

**1 Comment**: How confident are you in the aerosol assumptions used as input to your model? Aerosol chemistry is relevant when certain aerosols activate as CCN. Also, emission data is important since the Tibetan Plateau does not often witness heavy aerosol loading events, though you are correct about the summer season having the most polluted conditions. Also, coarse mode aerosols (mineral dust) tend to dominate that region throughout the year.

**Response:** We have clarified in Section 2: "*It is worth noting that the simple aerosol assumption is subject to cause rather large uncertainties in the aerosol activation to CCN and IN. Aerosol chemistry in clouds plays a considerable role in the aerosol nucleation and growth. Direct emissions from anthropogenic sources contribute substantially to the CCN and IN, even over the TP with increasing human activities. Furthermore, mineral dust from the natural source frequently dominates the TP throughout the year. Therefore, future studies need to be conducted to include all the aerosol modes, chemistry and emissions.*"

**2 Comment**: Do you expect to see much sulfate in the Tibetan Plateau? And if so, how does that compare to the ability of carbonaceous aerosols to activate as CCN since they too can be observed over this region.

**Response:** We have clarified in Section 2: "*Observed aerosol concentrations over the TP exhibit a large variation during the monsoon season, i.e., the observed sulfate concentrations range from 0.1 to several μg m$^{-3}$ (Decesari et al., 2010). Therefore, a set of 28 initial aerosol size distributions with the aerosol number concentration ranging from 20 to 9000 cm$^{-3}$ and the*

*sulfate mass concentration ranging from 0.02 to 9.0 μg cm⁻³ at the surface level are used. Other aerosol species are scaled using the measurement at the Nepal Climate Observatory-Pyramid (NCO-P) (Decesari et al., 2010). These aerosol distributions are designated for environments ranging from very clean background air mass to polluted urban plumes over the TP and NCP. Although the observed organic aerosol dominates the aerosol composition at NCO-P (Decesari et al., 2010), considering the large uncertainties in the hygroscopicity of organic aerosols, the hygroscopicity parameter for the secondary organic aerosol is set to be 0.05 in the study (Petters and Kreidenweis, 2007; 2008). Hence, sulfate aerosols (or inorganic aerosols) still play a dominant role in the CCN activation.".*

**3 Comment**: I have no problems with the results you show from your model but both comparison regions should be equivalent. It seems that topography plays a large role in the development of deep convection over the Tibetan Plateau. I believe your point is that the monsoon may be affected by aerosol-cloud interactions over this region. However, is it a fair to compare aerosol impacts on storm development in the Tibetan Plateau region with the Northern China Plains where the topography is not only flatter but there are more sources of aerosol loading as well.

**Response:** We agree with the reviewer's comment and have clarified in Section 4: "*It is worth noting that, although the CAPE is similar for the Cu-TP and Cu-NCP, it might not be fair to compare aerosol impacts on the cloud development over the TP with the NCP, considering the difference of the water vapor profile, wind shear, topography, and anthropogenic and natural aerosol sources between the two regions. However, the comparisons have highlighted that the topography plays a large role in the development of cumulus over the TP.*".

**4 Comment**: Please also note the supplement to this comment: http://www.atmos-chem-phys-discuss.net/acp-2017-148/acp-2017-148-RC2- supplement.pdf

**Response:** We really appreciate the reviewer for the careful reading and revisions of the manuscript and have revised it as suggested.